# Examining the relationships between early childhood experiences and adolescent and young adult health status in a resource-limited population: A cohort study

Zeba A. Rasmussen[1]*, Wasiat H. Shah[2], Chelsea L. Hansen[1], Syed Iqbal Azam[3], Ejaz Hussain[4], Barbara A. Schaefer[5], Nicole Zhong[5¤a], Alexandra F. Jamison[1], Khalil Ahmed[6], Benjamin J. J. McCormick[1], for the Oshikhandass Water, Sanitation, Health and Hygiene Interventions Project

1 Division of International Epidemiology and Population Studies, Fogarty International Center, National Institutes of Health, Bethesda, Maryland, United States of America, 2 Department of Medicine, Aga Khan University, Karachi, Pakistan, 3 Department of Community Health Sciences, Aga Khan University, Karachi, Pakistan, 4 Administration Department, Karakoram International University, Gilgit, Pakistan, 5 Department of Educational Psychology, Counseling, and Special Education, Pennsylvania State University, University Park, Pennsylvania, United States of America, 6 Faculty of Life Sciences, Karakoram International University, Gilgit, Pakistan

☯ These authors contributed equally to this work.
¤a Current address: Department of Human Services, School of Education and Human Development, University of Virginia, Charlottesville, Virginia, United States of America
* zeba.rasmussen@nih.gov

**Data Availability Statement:** Data is held in a public repository entitled the National Institute of Child Health and Human Development Data and

## Abstract

### Background

Adolescence is a critical point in the realization of human capital, as health and educational decisions with long-term impacts are made. We examined the role of early childhood experiences on health, cognitive abilities, and educational outcomes of adolescents followed up from a longitudinal cohort study in Pakistan, hypothesizing that early childhood experiences reflecting poverty would manifest in reduced health and development in adolescence.

### Methods and findings

Adolescents/young adults previously followed as children aged under 5 years were interviewed. Childhood data were available on diarrhea, pneumonia, and parental/household characteristics. New data were collected on health, anthropometry, education, employment, and languages spoken; nonverbal reasoning was assessed. A multivariable Bayesian network was constructed to explore structural relationships between variables. Of 1,868 children originally enrolled, 1,463 (78.3%) were interviewed as adolescents (range 16.0–29.3 years, mean age 22.6 years); 945 (65%) lived in Oshikhandass. While 1,031 (70.5%) of their mothers and 440 (30.1%) of their fathers had received no formal education, adolescents reported a mean of 11.1 years of education. Childhood diarrhea (calculated as episodes/child-year) had no association with nonverbal reasoning score (an arc was supported in just 4.6% of bootstrap samples), health measures (with BMI, 1% of bootstrap samples;

Specimen Hub (DASH) at https://dash.nichd.nih.
gov/study/416387.

**Funding:** The study was funded by the Pakistan US
S&T Cooperative Agreement (https://sites.
nationalacademies.org/pga/pakistan/index.htm)
between the Pakistan Higher Education
Commission (HEC, https://www.hec.gov.pk/
english/pages/home.aspx) (No.4-421/PAK-US/
HEC/2010/955, grant to the Karakoram
International University, to KA) and US National
Academies of Science (https://www.
nationalacademies.org/) (Grant Number PGA-
P211012 from NAS to the Fogarty International
Center, to ZAR). Original study funding: The
Applied Diarrheal Disease Research Program at
Harvard Institute for International Development
(Grants 063 and P033 to ZAR), and the Aga Khan
Health Service, Northern Areas and Chitral,
Pakistan (https://www.akdn.org/aga-khan-health-
service-pakistan-0, in kind support to ZAR). The
funders had no role in study design, data collection
and analysis, decision to publish, or preparation of
the manuscript.

**Competing interests:** The authors have declared
that no competing interests exist.

**Abbreviations:** BMI, body mass index; BN,
Bayesian network; G-B, Gilgit-Baltistan; HAZ,
length/height-for-age $z$ score; HCI, Human Capital
Index; SRPH, self-reported past childhood health;
SRCH, self-reported current health.

systolic and diastolic blood pressure, 0.1% and 1.6% of bootstrap samples, respectively),
education (0.7% of bootstrap samples), or employment (0% of bootstrap samples). Rela-
tionships were found between nonverbal reasoning and adolescent height (arc supported in
63% of bootstrap samples), age (84%), educational attainment (100%), and speaking
English (100%); speaking English was linked to the childhood home environment, mediated
through maternal education and primary language. Speaking English ($n$ = 390, 26.7% of
adolescents) was associated with education (100% of bootstrap samples), self-reported
child health (82%), current location (85%) and variables describing childhood socioeco-
nomic status. The main limitations of this study were the lack of parental data to characterize
the home setting (including parental mental and physical health, and female empowerment)
and reliance on self-reporting of health status.

## Conclusions

In this population, investments in education, especially for females, are associated with an
increase in human capital. Against the backdrop of substantial societal change, with the
exception of a small and indirect association between childhood malnutrition and cognitive
scores, educational opportunities and cultural language groups have stronger associations
with aspects of human capital than childhood morbidity.

## Author summary

### Why was this study done?

- Childhood diarrhea and pneumonia remain leading causes of morbidity and mortality,
  and their effects are hypothesized to have long-lasting impacts that could potentially
  negatively affect adolescent/young adult nutrition, reasoning skills, and educational
  status.

- Studies across different geographical settings have focused on outcomes during early
  childhood and on nutritional status or interventions, though very few have extended
  into adolescence/young adulthood.

- Other studies have noted negative associations between poverty, morbidity, and malnu-
  trition and childhood attainment and physical and cognitive development.

### What did the researchers do and find?

- We conducted a prospective longitudinal study on a cohort of children ($n$ = 1,868,
  1989–1996) from a poor, remote rural Pakistani village who were followed up 20 years
  later ($n$ = 1,463, 2011–2014), when they were adolescents and young adults (mean age
  22.6 years old).

- Most of their mothers (70.5%) and fathers (30.1%) had received no formal education,
  but the adolescents had, on average, 11.1 years of education.

- Childhood illness was not related to nonverbal reasoning score, health measures, education, or employment.

- Education (including speaking English) was strongly related to nonverbal reasoning score.

- Speaking English was related to improved childhood socioeconomic status, better child health, more years of education by adolescence, and greater likelihood of having left the village.

### What do these findings mean?

- Investment in education, especially for females, is important in the development of human capital.

- Childhood undernutrition was related to adolescent height, and thereby weakly to cognitive score, whereas educational opportunities and cultural language group were more strongly associated with human capital than childhood illness was.

- This population underwent substantial secular changes, including economic growth and expanded educational opportunities, that may have overcome deficits from childhood undernutrition.

## Introduction

Human capital is the knowledge, skills, and health that individuals accumulate over time that enables them to realize their potential as productive members of society [1]. The World Bank created in 2018 a population-level indicator of human capital by indexing child survival and stunting, school attendance and performance, and adult survival. Although essentially an economic construct, its goal is to capture shortfalls in the realization of human development. The Human Capital Index (HCI) describes how improvements to childhood health, nutrition, and early learning lay strong foundations for the future acquisition of cognitive, social, and behavioral skills [2]. As such, the adolescent period is a critical stage between childhood and adulthood where formal education ends and key health decisions are made that have long-term impacts [3,4].

Pakistan ranks 134 of 157 countries in the 2018 World Bank HCI [1]. Located in remote, mountainous northeast Pakistan, Gilgit-Baltistan (G-B) in the 1980s was one of the poorest regions of the country, with a per capita income less than half that of the rest of Pakistan [5], with subsistence farming as the primary occupation, low educational levels, low life expectancy (53 years), and high maternal mortality [6]. Considerable investment in rural support, education, and health changed this situation after the 1978 opening of the Karakoram Highway, which provided increased accessibility to the area, leading to substantial societal change [5]. Communities in G-B were strongly motivated to invest in children's education for future job opportunities and income [7]. The most recent evaluation in 2010 showed a dramatic secular transformation, resulting in a narrowing of the income gap, to a per capita income of about 90% of the national average [8].

We report the findings of an observational longitudinal study of a cohort of children who were followed up 15–20 years later, when they were adolescents and young adults (hereafter referred to as adolescents) [4]. The childhood cohort came from Oshikhandass, an ethnically mixed, remote rural village with high morbidity and mortality [9], 20 km from the capital of G-B. This study examines individual-level associations between childhood health and family socioeconomic indicators of early life adversity and adolescent health and development. Our study considers the following as outcomes of interest: nonverbal reasoning score, self-reported health status, body mass index (BMI), blood pressure, educational attainment, and employment. We hypothesized that early life experiences reflecting poverty would manifest in reduced health and development in adolescence, which are the foundation of adult human capital. [10].

## Methods

This study is reported as per the Strengthening the Reporting of Observational Studies in Epidemiology (STROBE) guidelines (see S1 Appendix) [11]. Prospective protocols (see S2 and S5 Appendices) were used for the collection of data but not for analysis.

### Ethics approval and consent to participate

Ethical approval was granted for the 1989–1996 study by the Aga Khan University Human Subjects Protection Committee (15 November 1989 for childhood diarrhea; 3 November 1993 for pneumonia); parents/legal guardians of children provided oral informed consent to participate, given high levels of parental illiteracy. Ethical approval for the adolescent study was granted by the US National Institutes of Health Institutional Review Board (#20TWN071, initially approved by the National Institute of Child Health and Development's institutional review board), the Aga Khan University Ethics Review Committee (1966-CHS-ERC-11), and Karakoram International University Ethics Review Committee. For the adolescent study, all participants aged 18 years or over provided signed consent to participate; participants under age 18 years provided assent, and their parents provided written consent for their participation.

### Data collection for children under 5 years, 1989–1996

From 1989 to 1996, all children under age 5 years in Oshikhandass were enrolled with parental consent in the prospective cohort of the Oshikhandass Diarrhea and Pneumonia Project. Local health workers visited families weekly until the child's fifth birthday, out-migration, or death; data collection methods and variable definitions are described elsewhere [9]. These data were used to estimate the burden of child morbidity (number of diarrhea and pneumonia episodes divided by days of follow-up). Pneumonia was not included in the main analyses because of a smaller sample size [9]. The mean length/height-for-age $z$ score (HAZ) was calculated for each child from their available quarterly data, using 2006 WHO growth standards. At enrollment, a survey was conducted of household characteristics (e.g., house type based on construction materials, number of rooms and occupants, and type of water and sanitation) and parental educational attainment (illiterate/no formal education, primary to 10 years, or more), occupation, and income [9].

### Data collection for follow-up in adolescence and young adulthood, 2011–2014

In July 2011, a village census identified participants from the original cohort who still lived in Oshikhandass, had migrated elsewhere, or died. Cohort members were contacted through their families. After obtaining written informed consent from adolescents, interviews were

conducted in Urdu and, rarely, in the participant's mother tongue (usually Burushaski or Shina), by study staff in person at the study office, at the participant's residence, or by telephone (<1%).

A questionnaire (see S3 Appendix) was used to collect data on current and past health status. The questionnaire was specifically designed for this study, considering important domains identified in the literature and in consultation with local subject specialists. The questionnaire was translated into Urdu, back-translated, and piloted in a neighboring village. Self-reported current health (SRCH) and self-reported past childhood health (SRPH) (between ages 5 and 15 years) were characterized using a 5-point scale (very poor to excellent), and other lifetime health problems were recorded as "any" versus "none." The questionnaire recorded educational attainment (years of schooling), whether classes were repeated, employment (student, employed, or not employed), marital status, number of children, current geographic location, age when moved, and languages spoken. Primary language (i.e., mother tongue) was defined as Burushaski or Shina; almost all respondents spoke Urdu, the national language.

The Raven's Standard Progressive Matrices and Colored Progressive Matrices (PsychCorp) [12] were administered (see S5 Appendix) in a quiet location without distractions to assess nonverbal reasoning skills, which are a critical component of cognitive ability [13]. The Raven's matrices are widely used internationally because they have minimal language requirements, thus minimizing the impact of cultural and linguistic differences [12]. Scores were calculated by adding the number of correct items and then scaled into a T score (mean 50, standard deviation 10). Scores were analyzed using exploratory and confirmatory factor analytic approaches to ensure construct validity [14] (see S4 Appendix).

Clinical measurements, taken once each by trained master's level field coordinators, included participant weight (Seca 872 digital scales) without shoes and in light clothing, height (HM200P PortStad Portable Stadiometer, Charder), and abdominal girth (tape measure). BMI was calculated, analyzed as a continuous variable, and for context summarized as underweight ($<18.5$ kg/m$^2$), normal (18.5–24.9 kg/m$^2$), overweight (25.0–29.9 kg/m$^2$), and obese ($\geq30$ kg/m$^2$); pregnant women were excluded. Blood pressure (in mm Hg) was measured once while the rested participant was sitting cross-legged on the floor using a mercury sphygmomanometer (Yamasu Model 600, Japan) and defined as normal except if elevated (systolic blood pressure 121–130 mm Hg and diastolic blood pressure $\leq 80$ mm Hg), stage 1 hypertension (systolic blood pressure 131–140 mm Hg or diastolic blood pressure 81–90 mm Hg), or stage 2 hypertension (systolic blood pressure $> 140$ mm Hg or diastolic blood pressure $> 90$ mm Hg). Missing data were excluded from the analysis described below; for example, clinical measurements and Raven's matrices were not available for adolescents interviewed by telephone.

## Data analysis

Univariate linear models were constructed using the Raven's score (squared to approximate normality) as the outcome. Subsequently, a multivariate Bayesian network (BN, using the R package bnlearn) [15] was constructed to explore the structural relationships between variables; BN analysis determines an optimal statistical description of all the data as it considers many possible model structures [16]. The best-fitting structure to describe the data was sought using a tabu greedy search [17], assuming either linear regression for continuous nodes or conditional probabilities for categorical (including binary) nodes, and a ban list to exclude directed arcs from any variable to the age and sex of the individual (biologically implausible) or from descriptions of their current health, education, or employment status to the variables characterizing their childhood environment (temporally impossible). The single best-fitting model, optimizing the Bayesian information criterion, was bootstrapped (7,500 times), and

only arcs present in 50% or more of samples were retained. To estimate confidence intervals for the primary outcomes, regression models were separately constructed using subnetworks identified from the BN.

All analyses were conducted in R (April 2018, release 3.5.0).

## Results

Of 1,868 children enrolled in the initial study, 1,463 (78.3%) were interviewed as adolescents (Fig 1). Almost two-thirds (*n* = 945) were still living in Oshikhandass, 35.1% (347) elsewhere within Pakistan, and 0.3% (5) abroad. Seventy-nine percent had complete information to build the BN (*n* = 1,165). Compared to the 1,462 individuals who were re-enrolled as adolescents, individuals lost to follow-up (*n* = 381) tended to be male (57.2% versus 51.0%, *p* = 0.017, chi-squared test) and to have more diarrheal episodes per child-year (mean [95% CI] 0.54 [0.00, 1.42] versus 0.42 [0.00, 1.02], *p* = 0.017, Kruskal–Wallis test), less improved household construction (62.2% improved or somewhat improved versus 65.7%, *p* = 0.002, chi-squared test), and more likely to have improved toilets (3.9% versus 2.3%, *p* = 0.032, chi-squared test) (see S1 Table). Of 126 (6.8%) children with follow-up who had died, 95 died during the first study and 31 died between studies. Those who did not complete the Raven's matrices (*n* = 60/1,462, 4.1%, most because they reported a lack of time), compared to those who did complete the Raven's matrices, were more likely to live away from Oshikhandass (63.3% versus 34.2%, *p* < 0.001, chi-squared test), were older (mean 26.7 versus 22.2 years old, *p* < 0.001, Kruskal–Wallis test), had a higher BMI (mean 23.0 versus 21.7 kg/m$^2$, *p* = 0.002, *t*-test), had fewer years of education (mean 7.9 versus 11.2 years, *p* < 0.001, *t*-test), and were less likely to speak English (16.7% versus 27.1%, *p* = 0.10, chi-squared test).

Selected cohort characteristics are shown in Table 1. Males outnumbered females in both the original and adolescent cohorts, despite more male deaths (71 male versus 55 female). Mean age at follow-up was 22.6 years (range 16.0–29.3) based on recorded birthdate. Only 14.4% were married; however, this percentage was higher among the female participants, who were married longer (mean 3.8 versus 2.2 years, *p* < 0.001, *t*-test), were married at a younger age (mean 21.6 versus 24.7 years, *p* < 0.001, *t*-test), and had more children (mean 1.8 versus 1.3, *p* = 0.01, *t*-test). At the time of interview, 54 women (7.5%) were pregnant. Females were more likely to remain in Oshikhandass, whereas males were more likely to move for education (30.6% versus 9.6%, *p* < 0.001, test of proportions) or employment (8.4% versus 0.8%, *p* < 0.001, test of proportions). In total, 1,240 (67.3%) adolescents had at least 1 childhood HAZ observation (median 8.0, IQR 9), with a population mean HAZ of −2.2, but the mean adolescent HAZ (assuming an age of 19 years, since WHO growth reference standards for calculating age- and sex-standardized HAZ only go up to this age) was −1 (see S1 Fig). Childhood HAZ was positively correlated with adolescent height (Pearson's rho 0.15 for females and 0.24 for males). Along with Urdu, almost all adolescents spoke Shina (1,414, 96.7%), the language most widely spoken in Gilgit, and about two-thirds also spoke Burushaski, the mother tongue of those originally from Hunza. A quarter (367, 25.1%) of adolescents spoke ≥4 languages, and the fourth most common language was English, spoken more often by males. In our population, based on a linear regression adjusting for age and sex, speaking English was associated with having approximately 2.4 more years of education (95% CI 2.1 to 2.7, *p* < 0.001); all English speakers had ≥8 years of education.

### Nonverbal reasoning score using the Raven's matrices

In univariate linear regressions (see S2 Table), the Raven's score (squared T score) was significantly (*p* ≤ 0.05) negatively related to childhood household density (−160 [95% CI −220, −91],

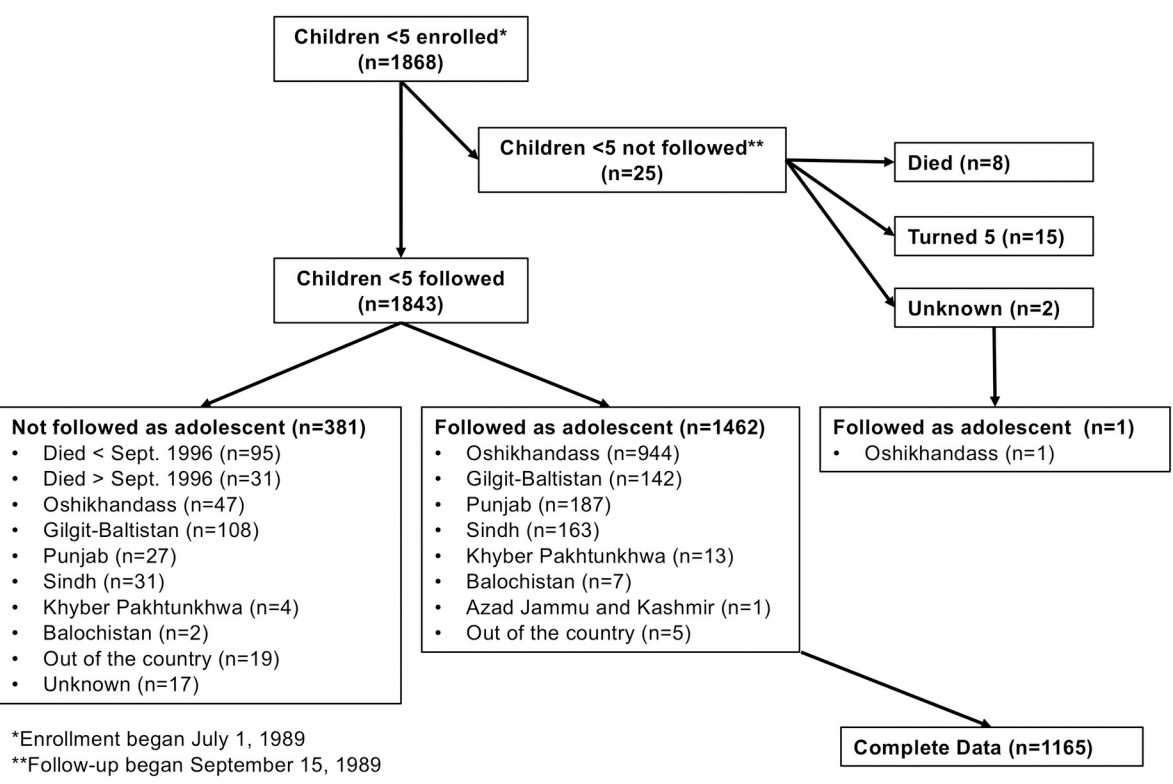

**Fig 1. Flowchart of participant follow-up.**

$p < 0.001$), inferior household (−400 [95% CI −530, −260], $p < 0.001$) or toilet construction (−650 [95% CI −980, −320], $p = 0.003$), lower status paternal occupation (−420 [95% CI −550, −280], $p < 0.001$), whether they were a Shina speaker (−440 [95% CI −550, −340], $p < 0.001$), and reported worse health (SRCH, −440 [95% CI −770, −370], $p < 0.001$; SRPH, −660 [95% CI −850, −460], $p < 0.001$); scores were positively related to being male (280 [95% CI 180, 370], $p < 0.001$), height (21 [95% CI 16, 26], $p < 0.001$), weight (13 [95% CI 7.6, 18], $p < 0.001$), waist girth (5.2 [95% CI 0.11, 10], $p = 0.045$), moving and living away from Oshikhandass (730 [95% CI 560, 900], $p < 0.001$), age at moving (408 [95% CI 249, 567], $p < 0.001$), whether they spoke English (850 [95% CI 750, 950], $p < 0.001$), being unmarried (350 [95% CI 200, 490], $p < 0.001$), being employed or a student (380 [95% CI 220, 540] and 550 [95% CI 410, 690], respectively, $p < 0.001$), having more years of education (130 [95% CI 110, 140], $p < 0.001$), not having repeated any classes (150 [95% CI 55, 250], $p = 0.002$), parents' educational status (maternal, 870 [95% CI 580, 1,200], $p < 0.001$; paternal, 610 [95% CI 470, 750], $p < 0.001$), maternal occupation (occupation other than housewife, 710 [95% CI 520, 900], $p < 0.001$), and higher parental income (320 [95% CI 150, 500], $p < 0.001$). The number of variables related to the Raven's score was substantially reduced in the multivariate BN, with T score directly associated with only an individual's adolescent height, age at assessment, educational attainment, and whether or not they spoke English (Fig 2). Older participants tended to have a lower Raven's score (−41.1 squared T score points per year [95% CI −55.8, −26.4], $p < 0.001$, linear regression), but taller participants and those with more years of education achieved higher scores (13.3 squared T score points per centimeter [95% CI 8.12, 18.5]; 91.2 squared T score points per year of schooling [95% CI 69.1, 113]; both $p < 0.001$; Fig 3).

Speaking English was strongly positively associated with the Raven's score as well as related to the current geographic location and SRPH. Participants who spoke English were more likely

**Table 1. Selected descriptive characteristics of the adolescent cohort.**

| Characteristic | Total | Male | Female | p-Value |
|---|---|---|---|---|
| Number of participants | 1,463 | 746 (51.0%); 746 (52.9%)* | 717 (49.0%); 663 (47.1%)* | 0.3[a] |
| Recorded age (years) | 22.6 (3.5) | 22.5 (3.4) | 22.6 (3.6) | 0.8[b] |
| Height (cm) | 162.9 (9.5) | 169.5 (7.0) | 156.0 (6.4) | <0.001[b] |
| Weight (kg)* | 57.6 (9.8) | 61.5 (9.5) | 52.9 (8.1)* | <0.001[b] |
| Waist girth (cm)* | 78.2 (9.6) | 79.8 (8.5) | 77.3 (10.1)* | <0.001[b] |
| BMI (kg/m$^2$)* | 21.7 (3.3) | 21.4 (3.1) | 21.8 (3.3)* | 0.02[b] |
| BMI category* | | | | 0.3[c] |
| Underweight (BMI < 18.5 kg/m$^2$) | 198 (13.5%) | 107 (14.3%) | 89 (13.4%)* | |
| Normal BMI (BMI 18.5–24.9 kg/m$^2$) | 1,065 (72.8%) | 555 (74.4%) | 477 (71.9%)* | |
| Overweight (BMI 25.0–29.9 kg/m$^2$) | 171 (11.7%) | 74 (9.9%) | 83 (12.5%)* | |
| Obese (BMI ≥ 30 kg/m$^2$) | 29 (2.0%) | 10 (1.3%) | 14 (2.1%)* | |
| Systolic BP (mm Hg) | 113.2 (10.2) | 115.2 (9.5) | 111.1 (10.4) | <0.001[b] |
| Diastolic BP (mm Hg) | 75.8 (8.5) | 77.0 (8.4) | 74.8 (8.4) | <0.001[b] |
| BP category** | | | | 0.002[c] |
| Normal BP | 1,234 (84.4%) | 605 (81.1%) | 629 (87.8%) | |
| Elevated BP | 42 (2.9%) | 26 (3.5%) | 16 (2.2%) | |
| Stage 1 hypertension | 157 (10.7%) | 93 (12.5%) | 64 (8.9%) | |
| Stage 2 hypertension | 29 (2.0%) | 22 (2.9%) | 7 (1.0%) | |
| Married | 211 (14.4%) | 41 (5.5%) | 170 (23.7%) | <0.001[a] |
| Raven's score (T score) | 50.0 (10.0) | 51.3 (10.0) | 48.6 (9.8) | <0.001[b] |
| Currently in Oshikhandass | 945 (64.6%) | 428 (57.4%) | 517 (72.1%) | <0.001[a] |
| Student*** | 984 (67.3%) | 515 (69.0%) | 469 (65.4%) | 0.2[a] |
| Employed*** | 269 (18.4%) | 223 (29.9%) | 73 (10.2%) | <0.001[a] |
| Highest level of education (years) | 11.1 (2.8) | 11.0 (2.7) | 11.2 (2.9) | 0.3[b] |
| Speaks English | 390 (26.7%) | 242 (32.4%) | 148 (20.6%) | <0.001[a] |
| Speaks Burushaski | 1,010 (69.0%) | 528 (70.8%) | 482 (67.2%) | 0.2[a] |
| Speaks Shina | 1,414 (96.7%) | 729 (97.7%) | 685 (95.5%) | 0.03[a] |
| Mother illiterate | 1,031 (70.5%) | 524 (70.5%) | 507 (71.2%) | >0.999[a] |
| Father illiterate | 440 (30.1%) | 236 (31.9%) | 204 (28.8%) | 0.75[a] |
| SRCH | | | | <0.001[c] |
| Excellent | 138 (9.4%) | 69 (9.2%) | 69 (9.6%) | |
| Good | 294 (20.1%) | 153 (20.5%) | 141 (19.7%) | |
| Satisfactory | 766 (52.4%) | 427 (57.2%) | 339 (47.3%) | |
| Poor | 262 (17.9%) | 96 (12.9%) | 166 (23.2%) | |
| Very poor | 3 (0.2%) | 1 (0.1%) | 2 (0.3%) | |
| SRPH | | | | 0.09[c] |
| Excellent | 117 (8.0%) | 57 (7.6%) | 60 (8.4%) | |
| Good | 388 (26.5%) | 194 (26.0%) | 194 (27.1%) | |
| Satisfactory | 561 (38.3%) | 308 (41.3%) | 253 (35.3%) | |
| Poor | 386 (26.4%) | 184 (24.7%) | 202 (28.2%) | |
| Very poor | 11 (0.8%) | 3 (0.4%) | 8 (1.1%) | |

Data are mean (SD) or *n* (%). BMI, body mass index; BP, blood pressure; SRCH, self-reported current health; SRPH, self-reported past childhood health.

*Fifty-four females were pregnant at the time of the interview. These individuals are not included in the weight, waist girth, and BMI measurements.

**Normal except if elevated (systolic blood pressure 121–130 mm Hg and diastolic blood pressure ≤ 80 mm Hg), stage 1 hypertension (systolic blood pressure 131–140 mm Hg or diastolic blood pressure 81–90 mm Hg), or stage 2 hypertension (systolic blood pressure > 140 mm Hg or diastolic blood pressure > 90 mm Hg).

***Categories are not mutually exclusive.

[a]Test of proportions.

[b]*t*-test for normally distributed data.

[c]Chi-squared test for categorical variables.

to have left Oshikhandass (63% [244/390] had moved away compared to 26% [274/1,071] of non-English speakers, $p < 0.001$, test of proportions) and to report excellent or good childhood health (39% [154/390] of those who spoke English versus 26% [278/1,071] of non-English speakers, $p < 0.001$, test of proportions). English was also related to participants' primary language: Burushaski speakers (35%, 286/812) were more likely to speak English than Shina speakers (8%, 29/344, $p < 0.001$, test of proportions). Speaking English was the only variable with robust evidence to link the childhood home environment (including parental and household characteristics, mediated through maternal education and primary language) and Raven's score. Adolescent participants who spoke English were more likely to have educated parents (41.1% versus 24.8%, $p = 0.024$, chi-squared test) and to have been raised in higher income households (37.0% versus 21.6%, $p = 0.034$, chi-squared test) and households with lower household density (2.6 versus 2.7 people/room, $p = 0.013$, Kruskal–Wallis test) (see S3 Table).

No relationships were identified in the BN for the number of episodes of childhood diarrhea, and no evidence emerged that either diarrhea or pneumonia was related to the major outcomes in separate regression models (see S4 and S5 Tables). In separate regressions, unemployed individuals tended to have significantly higher rates of childhood diarrhea than those employed (see S4 Table), but this association was not supported in the bootstrapped BN.

## Self-reported health status

Most participants reported satisfactory, good, or excellent health for both SRPH (72.8%, 849/1,165) and SRCH (81.9%, 957/1,165). However, females were more likely than males to report poor/very poor SRCH; this difference was less pronounced in the SRPH (Table 1). The single predictor of SRCH was SRPH (Fig 2). The few individuals who reported poor/very poor SRPH were also most likely to report poor SRCH (Fig 4).

Childhood health status was itself a predictor of lifetime health problems (Fig 2), with participants who characterized their childhood health as poor having an odds ratio of reporting later health problems of 3.4 (95% CI 2.3, 5.3, $p < 0.001$, multinomial regression; Fig 4). Lower SRPH (poor/satisfactory) was associated with repeated years of schooling.

## BMI

Height, weight, and waist girth were significantly higher among males than non-pregnant females, but BMI was slightly higher among females (Table 1). Additionally, the BN showed weak support (50% of bootstrap samples) for English-speaking individuals having a higher BMI (Fig 2).

## Education and employment

Participants had 11.1 years of education on average, and 984 (67.3%) were still studying. A higher proportion of males were employed (Table 1). Positive associations existed between years of education and speaking English and between education and employment (Fig 2). During childhood, most adolescents' mothers (1,031, 70.5%), and one-third (440, 30.1%) of fathers were illiterate (Table 1).

Most students (834, 85%) were full-time; a minority (88, 8.9%) were also employed. Of 479 non-students, 283 had completed studies, of whom 208 (73%) were employed, and the remaining 196 (85% of whom, or 166, were female) were doing unpaid family work. Employed participants ($n = 313$) described a variety of occupations, including teaching (42, 13.4%), business (35, 11.1%), and the army (25, 8.0%), but 106 (33.8%) had higher income roles (e.g., engineering, managerial/financial work, and hotel-related work). In contrast, from the childhood

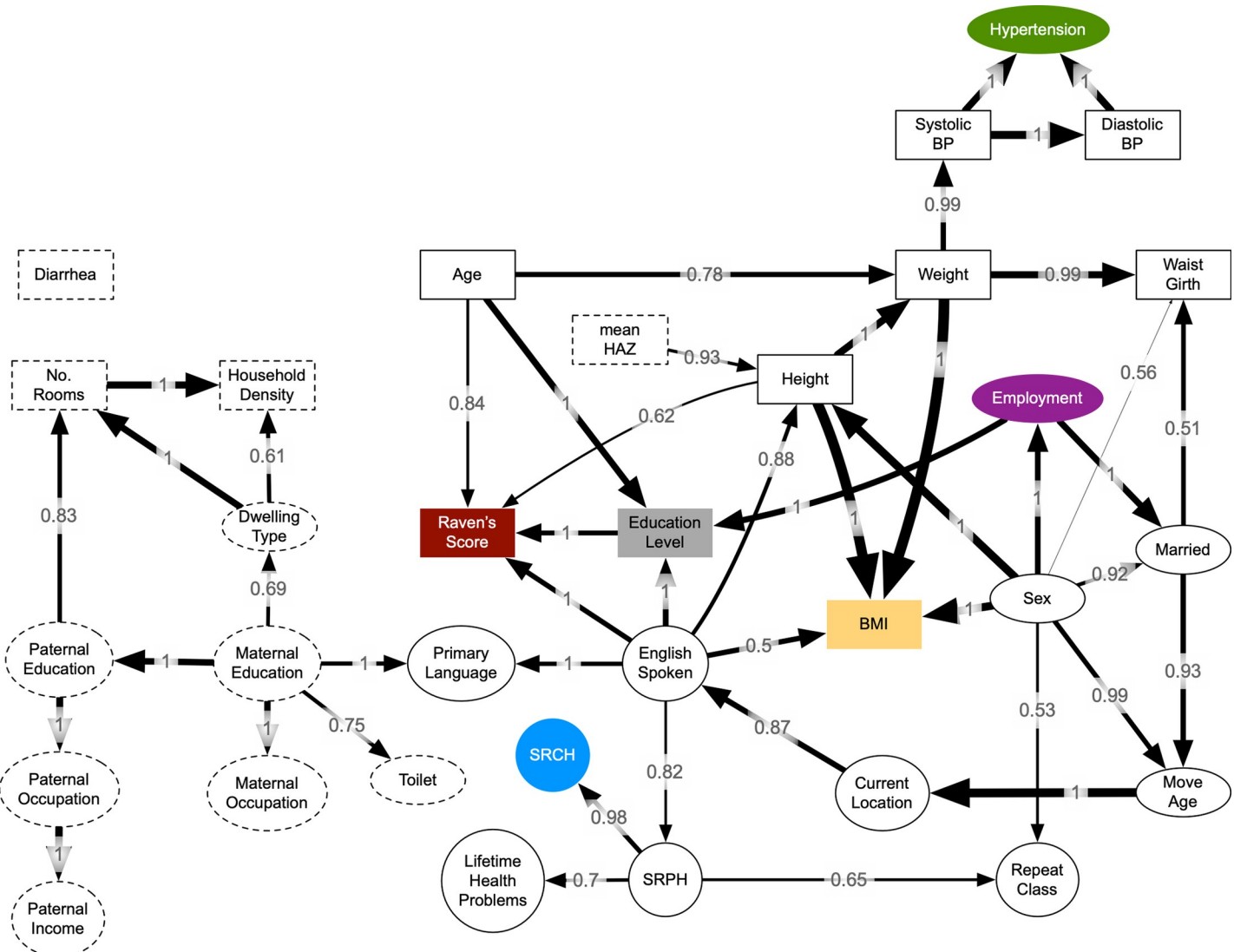

**Fig 2. Bayesian network showing associations that were robust in ≥50% of 7,500 bootstrap samples.** Continuous nodes are shown as rectangles, and categorical nodes as ellipses; childhood nodes are indicated by dashed outlines, and adolescent nodes by solid outlines. Arc width is proportional to the change in the Bayesian information criterion (BIC): Thicker arrows indicate a larger change in BIC and evidence of a more informative association. Numbers indicate the proportion of bootstrap iterations that supported a given arc. The main outcomes are highlighted (hypertension, Raven's score, SRCH, BMI, educational level, and employment status). The direction of associations is based on the maximum likelihood. BP, blood pressure; HAZ, length/height-for-age *z* score; No., number of; SRCH, self-reported current health; SRPH, self-reported past childhood health.

cohort study, their parents described more limited roles for mothers (724/806, 89.8%, were housewives) and for fathers (farming/labor, 35%; the army, 16%; and business, 16%).

## Discussion

Human capital is the capability of individuals to contribute to society, and depends, at least in part, on their health, education, employability, and cultural experiences. Adolescents are underrepresented in health research despite their unique role in representing the culmination of childhood life experiences and potentially predicting chronic illness in later life; most relevant literature has focused on the long-term consequences of early childhood malnutrition (see S6 Table). Adolescence also crucially captures the transition from child to actively

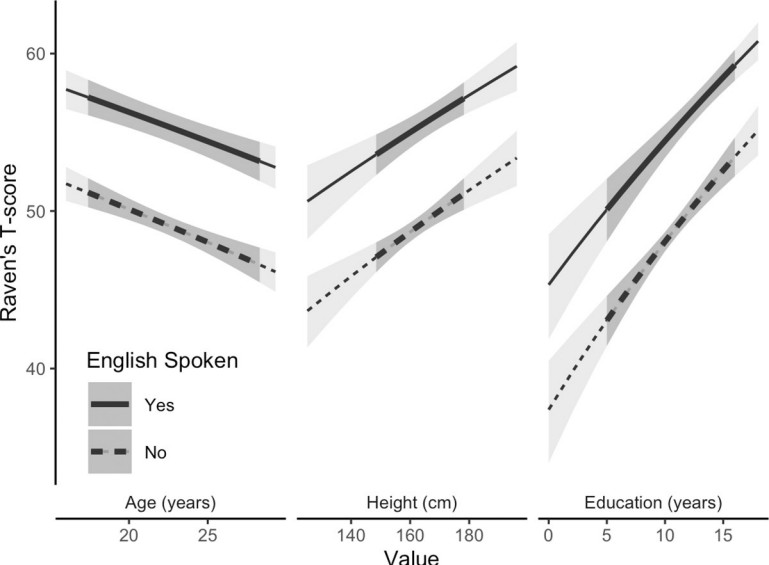

**Fig 3. Direct predictors of adolescent Raven's T score based on the structure identified in the Bayesian network.**
The Raven's score is a function of the participants' age, current height, their attained level of education, and whether they spoke English. The light grey 95% confidence intervals indicate the full range of each predictor, and the darker shading indicates the 95% confidence interval across the interquartile range of each respective predictor.

contributing adult, including becoming a parent. This study provides a unique opportunity to examine the linkages between childhood and adolescent health status. We hypothesized that the participants' early childhood experiences would affect the development and formation of human capital in this remote Pakistani village.

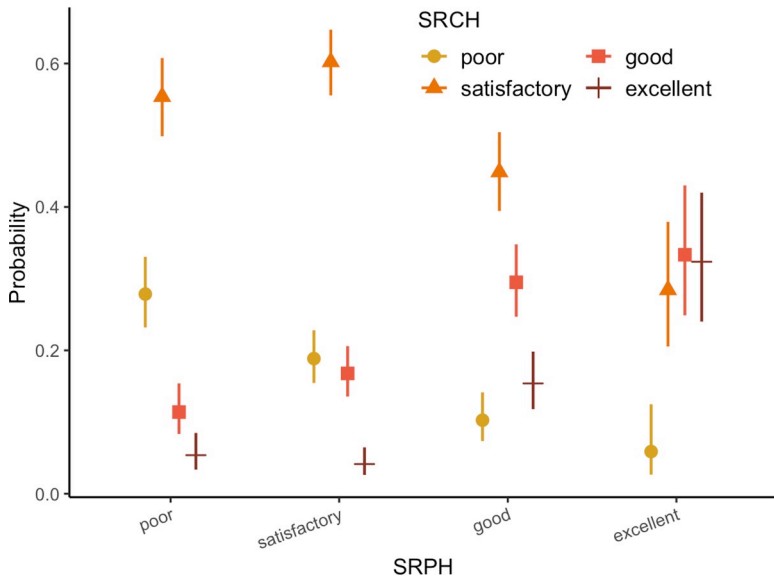

**Fig 4. The mean probability of the level of self-reported current health (SRCH) as an adolescent as a function of self-reported past childhood health (SRPH).** Whiskers indicate the 95% confidence intervals. Results based on the Bayesian network. The "very poor" category was pooled with the "poor" category due to the small number of responses.

Underpinning individual human capital potential is nonverbal reasoning. We found that, instead of childhood illness (i.e., diarrhea and pneumonia), nonverbal reasoning was more closely related to a collection of variables that shared an association with language, opportunities for education, and the home environment (e.g., maternal education and parental income). Additionally, we found some evidence to suggest that childhood malnutrition (indicated by HAZ) had a small indirect association with nonverbal reasoning. Two meta-analyses by Sudfeld et al. [18] and Victora et al. [19] reported positive associations between HAZ and cognitive scores, using cross-sectional and longitudinal data, respectively. Like the studies reanalyzed by Victora et al. [19], we found a positive association between childhood HAZ and later height; however, we did not find evidence of an association between undernutrition and cognitive development. Our results are consistent with findings that children who recover from stunting show no difference in cognitive outcomes from those who were not stunted [20–22]. This population underwent substantial secular changes, illustrated by the region's increases in per capita GDP, diversification of employment opportunities, and access to education. These factors may have alleviated some of the common constraints on both growth and cognitive development during an important period of development, but are also suggestive of opportunities to intervene that extend beyond early childhood [23].

In earlier analyses of a subset of this cohort ($n = 107$) [24], childhood morbidity was significantly associated with lower educational performance at 7 years old, consistent with the findings of other studies [25,26]. Education is a major component of the HCI. In 2019, the World Bank estimated that a Pakistani child born in 2018 can expect to complete 8.8 years of school, but only 4.8 years when adjusting for the quality of education [1]. In our population, the mean number of years of education was higher for both females and males, at 11.1, and was significantly higher than in the previous generation, where almost three-quarters of mothers and a third of fathers were illiterate or had no formal schooling. Although it is not possible to determine with these data, school performance, rather than attendance, may still be negatively impacted by childhood malnutrition and morbidity.

Childhood mortality in Oshikhandass decreased significantly from 1989 to 1996 [9], allowing families to focus more on helping surviving children grow and thrive. During this time, educational opportunities significantly improved in Oshikhandass, from there being 1 government high school for boys and 1 middle school for girls to the current situation of 9 early childhood education centers, 11 schools, and 2 colleges (up to class 12). One reason that the Burushaski speakers tended to have more years of education is that the leaders of the Ismaili community (primarily Burushaski speakers in Oshikhandass) emphasized its importance, especially for girls [7]. In Oshikhandass, there were major investments in education, especially improving opportunities for females. Notably, many adolescents in this cohort continued their education at university in Gilgit or elsewhere. Given the importance of this age in developing resilience, understanding how experiences and stresses influence development into adulthood is of great importance [27].

Our key hypothesis was that long-lasting associations between childhood health experiences and adolescent nonverbal reasoning and health status would be identified. We found no evidence for direct associations between the childhood and family factors measured and adolescent nonverbal reasoning. However, participants' mean childhood HAZ, which is an indicator of longer-term nutritional status, was positively correlated with adolescent height (approximately 1 cm per 1 $z$-score change, approximately a third of the association found in other studies [19]) and thereby indirectly to Raven's score (3 T score points per 1 cm adolescent height); this is similar to the findings of the other studies described above, but we did not find the strong associations others did. Our finding that neither childhood diarrhea nor pneumonia was associated with adolescent Raven's score contrasts to studies of cognitive scores in early

 

childhood that report negative impacts of nutrition, infection, and illness [28], but is consistent with findings that such early effects diminish by age 5 years [29]. Given the temporal gap between childhood illness, potential recovery from stunting, and the age at Raven's test administration, this is perhaps not surprising [30].

Previous studies have examined how the mesoenvironment [31], including a nurturing and stimulating home environment [10,32], becomes increasingly influential in child development as a child ages. Whether or not English was spoken by the participant was identified as a variable that linked both to Raven's score and to a suite of variables (parental education and occupation, paternal income, dwelling type, household density, type of toilet, and primary language) that characterized a participant's childhood home environment. We interpreted speaking English as an indication of higher childhood socioeconomic status and greater educational opportunity, as participants from lower socioeconomic backgrounds were less likely to speak English; speaking English was also associated with participants' opportunities to travel beyond Oshikhandass for either work or education.

The Raven's matrices are not typically thought to be biased by educational background and are designed to test observation, clarity of thinking, and nonverbal reasoning skills [12]. Here, we found that participants with more years of education tended to achieve higher Raven's scores. Other studies have similarly shown a positive correlation between educational age (up to a chronological age of 20 years) and Raven's score that might indicate a familiarity with the type of logic assessed, distinct from familiarity with the test per se [33].

SRCH was related to the participant's SRPH, but not to objective measures of health (e.g., BMI, anthropometry, or blood pressure). SRCH was also not related to socioeconomic variables, as might be expected based on the theory underpinning human capital [34]. However, self-reported health remains a meaningful tool and predictor of mortality [35].

BMI is one objective measurement of health with correlations to many chronic conditions that manifest in adulthood—such as diabetes, cancer, osteoarthritis, and cardiovascular diseases [36]—that influence adult survival and therefore the HCI. Blood pressure and hypertension are also objective measures of health, but here only related to weight; neither of these metrics was associated to childhood health except through indirect association with HAZ (via weight and height). Hypertension is reported to be a significant problem in northern Pakistan [37]. It is potentially related to salted tea consumption, and our finding of high rates of diastolic hypertension suggests that it will be important to monitor the prevalence of noncommunicable diseases [4,38] in the future in this area.

Childhood experiences varied by sociocultural group, indicated by the participant's primary language. The Burushaski subpopulation, as compared to the Shina subpopulation, were more likely speak English, to have moved out of the area, to reach higher educational levels, and to have come from families that had a higher socioeconomic status. Similarly, there were notable sex differences, with males tending to have a greater opportunity for continuing their education than females, as they were less likely to have married or to have children, and more likely to have traveled away from Oshikhandass for further study and employment. Additional targeted research would be required to better understand why adolescent females in Oshikhandass reported poorer perceived health, as reported elsewhere in Pakistan [39], and to understand the status of female empowerment.

Following the same cohort from birth to adolescence/young adulthood gives a rare opportunity to understand links from early life experiences to later outcomes, but it comes with the challenge of loss to follow-up. Given that this study was built upon an earlier study, the variables collected were also constrained by what was measured previously (e.g., no cognitive, blood pressure, micronutrient, dietary diversity, or food insecurity indicators were collected during childhood, nor were more detailed parental variables such as maternal mental health).

The age at enrollment and duration of initial follow-up also varied. Using the mean HAZ over available HAZ measurements was an attempt to mitigate the variable ages at observation; however, summarizing the data over the first 5 years of life obscures considerable development that takes place in early childhood. Additionally, as the original participants were followed up later, we used retrospective questions to ask about their perceived childhood health at the same time as their perceived current health status. This may lead to biased responses, especially recalling events over several years, and is acknowledged as a subjective rather than objective assessment of health, although we did include other clinical assessments of general health. Another limitation is measurement of school attendance, but not school performance, as has been used elsewhere [26], including in the HCI.

This study does have several strengths; principally, as a prospective longitudinal study spanning over 20 years, it adds significantly to the body of research examining the effects of childhood health and early life exposures on adolescent health status. Additionally, this study examined not only physiological health, but also reasoning abilities and self-perceptions of health.

In summary, we found that adolescents raised in a remote rural village in northeastern Pakistan had high rates of education—significantly higher than their parents—and that most were still students. Early childhood illness (diarrhea or pneumonia) was unrelated to Raven's score, current self-reported health status, BMI, hypertension, education, or employment. The primary language spoken did link childhood socioeconomic factors, characterized by maternal education, to adolescent outcomes. Sex differences were already apparent in these adolescent outcomes. We present evidence that there are opportunities to improve human capital beyond early childhood, and that with targeted investments such gains can be made even over a single generation.

## Supporting information

**S1 Fig. The distributions of HAZ from childhood measurements and approximated for adolescents.** Childhood measurements (dashed lines) and approximations for adolescents (solid lines). The childhood distribution is the mean value of each child's available measurements. Adolescents who were older than 19 years when their height was measured were assumed to be 19 years old because this is maximum age in the WHO growth reference to calculate the age- and sex-standardized HAZ. The black vertical lines indicate the mean childhood and adolescent HAZs. Of the 54.1% (630/1,025) of participants who were stunted as a child (HAZ < −2 SD from the median), 84.9% had recovered by adolescence (i.e., 535/630 had HAZ > −2), but 8.4% who had not been stunted as children became stunted as adolescents (45/535).
(DOCX)

**S1 Table. Selected descriptive characteristics comparing those lost to follow-up and those re-enrolled as adolescents.**
(DOCX)

**S2 Table. Univariate relationships predicting the squared Raven's T score assuming a linear regression between each variable and the squared Raven's T score.**
(DOCX)

**S3 Table. Characteristics of the cohort according to whether adolescents spoke English.**
(DOCX)

**S4 Table. Regression models of the primary outcomes and diarrheal episodes.** Showing coefficients from linear models for Raven's T score, BMI, and blood pressure; a Poisson model for education (log count); and a multinomial for employment status (log odds), all as a function of childhood diarrhea episodes. Variable inclusion was based on the Bayesian network.
(DOCX)

**S5 Table. Regression models of the primary outcomes and pneumonia episodes.** Showing coefficients from linear models for Raven's T score, BMI, and blood pressure; a Poisson model for education (log count); and a multinomial for employment status (log odds), all as a function of childhood pneumonia episodes. Variable inclusion was based on the Bayesian network.
(DOCX)

**S6 Table. Literature search.** Forty-five studies identified using 2 search strategies of the PubMed (US National Library of Medicine) database were manually filtered by the first and last author for relevance to this study. Search 1: (cohort OR longitudinal) AND (childhood OR adolescence) AND (cognitive OR cognition) AND (diarrhea OR diarrhoea OR pneumonia OR "childhood illness*") AND (BMI OR "health progression" OR growth OR "child development" OR "childhood development"). Search 1 yielded 21 studies. Search 2: ((child* OR infant OR adolescen*) AND (cohort* OR longitudinal OR followup OR "follow up")) AND cogniti* AND (("human capital" OR "human capacit*" OR potential OR "child development" OR "health progression" OR "adolescent development") AND ((adult* OR adoles*) AND outcome*))) AND (undernutrition OR stunt* OR diarrhea OR diarrhoea OR pneumonia OR "childhood illness*"). Search 2 yielded 34 studies. And 30 more studies were identified based on reviewers' comments.
(DOCX)

**S1 Appendix. STROBE checklist for observational studies.**
(DOCX)

**S2 Appendix. Protocol (adolescent study).**
(DOCX)

**S3 Appendix. Questionnaire (adolescent study).**
(PDF)

**S4 Appendix. Description of the psychometric analysis of the Raven's Standard Progressive Matrices and Colored Progressive Matrices.**
(DOCX)

**S5 Appendix. Protocol and scoring form for Raven's Standard Progressive Matrices and Colored Progressive Matrices administration.**
(DOCX)

## Acknowledgments

We are deeply indebted to the adolescents and their families who participated in this study and the dedicated project staff, including from Aga Khan University: Ahmed Jan and Saba Wasim; Fogarty International Center: Elizabeth Thomas, Faran Sikandar, Julia Baker, and Stacey Knobler; data collectors: Assis Jahan, Faheemullah Beg, Farah Naz Hashmani, Liaqat Ali, Rizwan Karim, and Shafiqa Yar Baig; Karakoram International University: Arif Hussain, Mirza Jibran, and Asif Hussain; study workers: Zohra Bano, Mobina Bano, Gul Nasreen, Mehtab Bano, Kaniz Fatima, Iqbal Bano, Dil Roz, Nazara, Ghazala, Nasima Begum, Alia Rani, Mehwish Hakeem, Rubina, Sameena, Zevar Jan, Sunaira, and Resham Jan. We thank Alicia

Livinski, National Institutes of Health Library, for literature searching, editing, and manuscript preparation assistance, and Professors Laura Murray-Kolb and Zulfiqar Bhutta and Drs. David J. Spiro, Peter Kilmarx, Cecile Viboud, and Roger Glass for editorial comments.

The findings and conclusions in this paper are those of the authors and do not necessarily represent the official position of the National Institutes of Health or US Department of Health and Human Services.

## Author Contributions

**Conceptualization:** Zeba A. Rasmussen, Chelsea L. Hansen, Syed Iqbal Azam, Khalil Ahmed, Benjamin J. J. McCormick.

**Data curation:** Zeba A. Rasmussen, Wasiat H. Shah, Chelsea L. Hansen, Syed Iqbal Azam, Ejaz Hussain, Barbara A. Schaefer, Alexandra F. Jamison, Benjamin J. J. McCormick.

**Formal analysis:** Zeba A. Rasmussen, Wasiat H. Shah, Chelsea L. Hansen, Syed Iqbal Azam, Barbara A. Schaefer, Nicole Zhong, Alexandra F. Jamison, Benjamin J. J. McCormick.

**Funding acquisition:** Zeba A. Rasmussen, Khalil Ahmed.

**Investigation:** Zeba A. Rasmussen, Wasiat H. Shah, Syed Iqbal Azam, Ejaz Hussain.

**Methodology:** Zeba A. Rasmussen, Wasiat H. Shah, Syed Iqbal Azam, Ejaz Hussain, Barbara A. Schaefer, Khalil Ahmed.

**Project administration:** Zeba A. Rasmussen, Wasiat H. Shah, Chelsea L. Hansen, Syed Iqbal Azam, Ejaz Hussain, Khalil Ahmed.

**Resources:** Zeba A. Rasmussen.

**Supervision:** Zeba A. Rasmussen, Wasiat H. Shah, Syed Iqbal Azam, Ejaz Hussain, Barbara A. Schaefer, Khalil Ahmed.

**Validation:** Zeba A. Rasmussen, Wasiat H. Shah, Syed Iqbal Azam, Ejaz Hussain.

**Visualization:** Zeba A. Rasmussen, Chelsea L. Hansen, Barbara A. Schaefer, Alexandra F. Jamison, Benjamin J. J. McCormick.

**Writing – original draft:** Zeba A. Rasmussen, Wasiat H. Shah, Chelsea L. Hansen, Benjamin J. J. McCormick.

**Writing – review & editing:** Zeba A. Rasmussen, Wasiat H. Shah, Chelsea L. Hansen, Syed Iqbal Azam, Ejaz Hussain, Barbara A. Schaefer, Nicole Zhong, Alexandra F. Jamison, Khalil Ahmed, Benjamin J. J. McCormick.

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
