## [Editor Report · Decision Letter 0]

1 Mar 2021

Dear Dr Rasmussen, 

Thank you for submitting your manuscript entitled "Unlocking human capital: revealing relationships between early childhood experiences and adolescent and young adult health status in a resource-limited population" for consideration by PLOS Medicine.

Your manuscript has now been evaluated by the PLOS Medicine editorial staff and I am writing to let you know that we would like to send your submission out for external peer review.

Please re-submit your manuscript within two working days, i.e. by March 4, 2021.

Kind regards,

Beryne Odeny

Associate Editor

PLOS Medicine

---

## [Decision Letter · Decision Letter 1]

10 May 2021

Dear Dr. Rasmussen,

Thank you very much for submitting your manuscript "Unlocking human capital: revealing relationships between early childhood experiences and adolescent and young adult health status in a resource-limited population" (PMEDICINE-D-21-00948R1) for consideration at PLOS Medicine. 

[LINK]

In light of these reviews, I am afraid that we will not be able to accept the manuscript for publication in the journal in its current form, but we would like to consider a revised version that addresses the reviewers' and editors' comments. Obviously we cannot make any decision about publication until we have seen the revised manuscript and your response, and we plan to seek re-review by one or more of the reviewers. 

We expect to receive your revised manuscript by May 31 2021 11:59PM. Please email us (plosmedicine@plos.org) if you have any questions or concerns.

We look forward to receiving your revised manuscript. 

Sincerely,

Beryne Odeny, 

PLOS Medicine

plosmedicine.org

Thank you for your submission. Before we proceed, please address the following editorial and reviewer comments.

1) Please revise your title according to PLOS Medicine's style. Your title must be nondeclarative and not a question. It should begin with main concept if possible. Please place the study design (e.g. "A prospective cohort study,") in the subtitle (i.e., after a colon). 

2) Abstract summary - At this stage, we ask that you reformat your non-technical Author Summary. The Author Summary should immediately follow the Abstract in your revised manuscript. This text is subject to editorial change and should be distinct from the scientific abstract. The summary should be accessible to a wide audience that includes both scientists and non-scientists. Please see our author guidelines for more information: https://journals.plos.org/plosmedicine/s/revising-your-manuscript#loc-author-summary.

3) In the abstract Methods and Findings:

a) Please ensure that all numbers presented in the abstract are present and identical to numbers presented in the main manuscript text.

b) Please quantify the main results with both 95% CIs and p values.

c) Please include the important dependent variables that are adjusted for in the analyses.

d) In the last sentence of the Abstract Methods and Findings section, please describe the main limitation(s) of the study's methodology.

4) Did your study have a prospective protocol or analysis plan? Please state this (either way) early in the Methods section.

5) Please add the following statement, or similar, to the Methods: "This study is reported as per the Strengthening the Reporting of Observational Studies in Epidemiology (STROBE) guideline (S1 Checklist)." 

6) Please include the completed STROBE checklist as Supporting Information. When completing the checklist, please use section and paragraph numbers, rather than page numbers.

7) In the methods, please describe how the adolescent questionnaire was developed and verified. Please provide.

8) If you developed a questionnaire as part of this study and it is not under a copyright more restrictive than CC-BY, please include a copy, in both the original language and English, as Supporting Information, or include a citation if it has been published previously.

9) In statistical methods, please refer to any post-hoc corrections to correct for multiple comparisons during your statistical analyses. If these were not performed please justify the reasons. Please refer to our statistical reporting guidelines for assistance (https://journals.plos.org/plosone/s/submission-guidelines.#loc-statistical-reporting)

10) In statistical methods, please discuss how you accounted for clustering of repeated measurements in this cohort.

11) Your study is observational and therefore causality cannot be inferred. Please remove language that implies causality, such as “greater influence on”, “have greatly increased”, or “investments in education have unlocked...” Refer to associations instead. Please temper the last sentence of your conclusion to avoid overreaching what can be concluded from the data. For example, refer to phrases such as “education has the potential to unlock…” and so forth.

12) In the Methods and Results section:

a) Please provide 95% CIs and p values for estimates in the main text and tables

b) When a p value is given, please specify the statistical test used to determine it.

13) Figures and tables:

a) Please indicate in the figure caption the meaning of the whiskers in Fig 4

b) Please define the following abbreviations in your tables. For example, IQR, SRCH, HAZ, BP, BMI

14) The terms gender and sex are not interchangeable (as discussed in http://www.who.int/gender/whatisgender/en/ ); please use the appropriate term.

15) Please use the "Vancouver" style for reference formatting and see our website for other reference guidelines https://journals.plos.org/plosmedicine/s/submission-guidelines#loc-references.

Comments from the reviewers:

Reviewer #1: See attachment

Michael Dewey

Reviewer #2: Rasmussen and colleagues use longitudinal data show that investments in education of females is associated with adult human capital in a Pakistani village decades later. Research of merit and has broad implications. A few points.

1.Secular trends - How was this addressed in this work. Many populations improve in health/well being over time(in fact from S1, your cohort got healthier between childhood and adolescence). I am curious of how was these shifts were addressed in your analyses and how if might impact your overall findings. 

2.The role of undernutrition in early life it seems to me is under-emphasized by authors. We know growth, brain development and other systems can all impact outcomes later in life -- all need proper nutrition. Similar work in Guatemalan birth cohort, and other LMICs showed that nutritional investments in females had impact on adult human capital and intergenerational benefits, with regards to offspring birth size. Underweight children have been found to be associated with failing a school grade in early life. It would be good if these points are discussed in this work to improve its contributory value.

3. Measurements - What was done to ensure reliability of clinical measurements(WT, HT, BP, WC etc) ? 

4. Raven score tests were not given to adolescents who had a phone interview. What proportion of the of your sample were these? How different were completers vs those who weren't interviewed?

Reviewer #3: The study tries to examine the role of early childhood experiences on health, cognition, and education outcomes of adolescents/young adults in Gilgit Baltistan, Pakistan. Childhood data collected was compared with data collected in adulthood to examine the relationship. 

Strong introduction and methods sections. As the authors also highlight in line 64/65 that female empowerment and mental health variables were not available, yet the result section provides interesting insights. 

If available, reference number 5 needs to be updated to a recent study/report. 

Reference numbers for the ethics approval need to be reported in ethics approval section. 

Although authors discuss bias in lines 423-424, some discussion on recall bias needs to be reported as a limitation for SRPH.

The manuscript has the potential to add to literature on examining the relationship of early and childhood health on adolescent/young adult health outcomes/status. 

Reviewer #4: This well-structured manuscript describes an important study which investigated the non-verbal cognitive outcomes of a group of adolescents and young adults followed up following an early life cohort study of under-5 mortality and morbidity in Pakistan. This study leverages the value created by a large initial cohort study from the 1990's and spans of period of rapid socio-economic mobility due to increased accessibility of one of the poorest regions in Pakistan. While the high level results reported here of higher SES (in particular, maternal education), predicting better cognitive outcomes are not highly novel, they are nevertheless important given the region on which the study reports, the authors do include discussion on the context specific elements at a more granular level which are highly relevant and important in thinking about potential of such findings to inform policy. 

The inclusion of measures of physical health (growth metrics, BMI, hypertension) as signals of potential risk for future NCD's is an additional strength. The methods and analysis approach is well described including specific reasons for children lost to follow up. 

As noted in the abstract, measures on maternal mental health does seem like an important limitation and the potential role that this may have played deserves some discussion in the limitations section. Further discussion on the critical role this period of life represents in terms of being the crucible for intergenerational risk and resilience would add value.

No major revisions needed from my perspective, though suggest a formal statistical review and publication of analysis plan if possible

[LINK]

---

## [Decision Letter · Decision Letter 2]

8 Jul 2021

Dear Dr. Rasmussen,

Thank you very much for re-submitting your manuscript "Examining the relationships between early childhood experiences and adolescent and young adult health status in a resource-limited population:  A prospective cohort study" (PMEDICINE-D-21-00948R2) for review by PLOS Medicine.

I have discussed the paper with my colleagues and the academic editor and it was also seen again by three reviewers. I am pleased to say that provided the remaining editorial and production issues are dealt with we are planning to accept the paper for publication in the journal.

[LINK]

We look forward to receiving the revised manuscript by Jul 15 2021 11:59PM.   

Sincerely,

Beryne Odeny, 

Associate Editor 

PLOS Medicine

plosmedicine.org

Requests from Editors:

1) Please remove the term “prospective” from the title as it is not clear that the research was prospectively planned, and there was no prespecified analysis plan. 

2) Adolescent participants provided “written informed consent” - would assent and parental consent not be more usual? Either way, please clarify under which circumstances they offered assent or consent (e.g., married, or emancipated minors consented, while assent and/or parental consent was sought for younger adolescents?)

3) For references #3 & #10, please provide access dates for the referenced weblinks. Please ensure that all weblinks are accessible and access dates updated.

Comments from Reviewers:

Reviewer #1: The authors have addressed all my points.

Michael Dewey

Reviewer #2: No further comments 

Reviewer #4: my comments have been adequately addressed by the authors

[LINK]

---

## [Editor Report · Decision Letter 3]

28 Jul 2021

Dear Dr Rasmussen, 

On behalf of my colleagues and the Academic Editor, Dr. Kathryn Mary Yount, I am pleased to inform you that we have agreed to publish your manuscript "Examining the relationships between early childhood experiences and adolescent and young adult health status in a resource-limited population:  A cohort study" (PMEDICINE-D-21-00948R3) in PLOS Medicine.

PRESS

Sincerely, 

Beryne Odeny 

Associate Editor 

PLOS Medicine